# NOVEL POLICY SEEKING WITH CONSTRAINED OPTIMIZATION

## ABSTRACT

We address the problem of seeking novel policies in reinforcement learning tasks. Instead of following the multi-objective framework commonly used in existing methods, we propose to rethink the problem under a novel perspective of constrained optimization. We first introduce a new metric to evaluate the difference between policies, and then design two practical novel policy seeking methods following the new perspective, namely the Constrained Task Novel Bisector (CTNB), and the Interior Policy Differentiation (IPD), corresponding to the feasible direction method and the interior point method commonly known in the constrained optimization literature. Experimental comparisons on the MuJuCo control suite show our methods can achieve substantial improvement over previous novelty-seeking methods in terms of both the novelty of policies and their performances in the primal task.[1]

## 1 INTRODUCTION

In Reinforcement Learning, an agent interacts with the environment to learn a policy that can maximize a certain form of cumulative rewards (Sutton & Barto, 1998), while the policy gradient method can be applied to optimize parametric policy functions (Sutton et al., 2000). However, direct optimization with respect to the reward function is prone to get stuck in sub-optimal solutions and therefore hinders the policy optimization (Liepins & Vose, 1991; Lehman & Stanley, 2011; Plappert et al., 2018). Consequently, an appropriate exploration strategy is crucial for the success of policy learning (Auer, 2002; Bellemare et al., 2016; Houthooft et al., 2016; Tang et al., 2017; Ostrovski et al., 2017; Tessler et al., 2019; Ciosek et al., 2019).

Recently many works have shown that incorporating curiosity in the policy learning leads to better exploration strategies (Pathak et al., 2017; Burda et al., 2018a;b; Liu et al., 2019). In these works, visiting a previously unseen or infrequent state is assigned with an extra curiosity bonus reward. Different from those curiosity-driven methods which focus on discovering new states within the learning procedure of a repeated single policy, the alternative approach of Novel Policy Seeking (Lehman & Stanley, 2011; Zhang et al., 2019; Pugh et al., 2016) focuses on learning different policies with diverse or the so-called novel behaviors to solve the primal task. In the process of novel policy seeking, policies in new iterations are usually encouraged to be different from previous policies. Therefore novel policy seeking can be viewed as an extrinsic curiosity-driven method at the level of policies, as well as an exploration strategy for a population of agents. Besides encouraging exploration (Eysenbach et al., 2018; Gangwani et al., 2018; Liu et al., 2017), novel policy seeking is also related to policy ensemble (Osband et al., 2018; 2016; Florensa et al., 2017) and evolution strategies (ES) (Salimans et al., 2017; Conti et al., 2018).

In this work, we aim at generating a set of policies that behave differently from all previous given policies while trying to keep their primal task performance. In order to generate novel policies, previous work often defines a heuristic metric for novelty estimation, e.g., differences of state distributions estimated by neural networks are used in (Zhang et al., 2019), and tries to solve the problem under the formulation of multi-objective optimization. However, most of these metrics suffer from the difficulty when dealing with episodic novelty reward, i.e., the difficulty of episodic credit assignment (Sutton et al., 1998), thus their effectiveness in learning novel policies is limited.

---

[1]Code will be made publicly available.

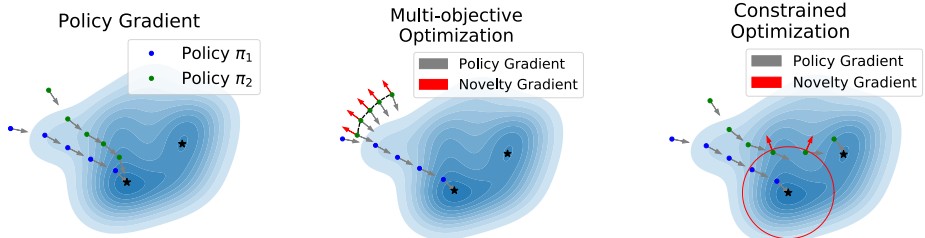

Figure 1: The comparison of the standard policy gradient method without novelty seeking (left), multi-objective optimization method (middle), and our constrained optimization approach (right) for novel policy seeking. The standard policy gradient method does not try actively to find novel solutions. The multi-objective optimization method may impede the learning procedure when the novelty gradient is being applied all the time (Zhang et al., 2019), e.g., a random initialized policy will be penalized from getting closer to the previous policy due to the conflict of gradients, which limits the learning efficiency and the final performance. On the contrary, the novelty gradient of our constrained optimization approach will only be considered within a certain region to keep the policy being optimized away from highly similar solutions. Such an approach is more flexible and includes the multi-objective optimization method as its special case.

Moreover, the difficulty of balancing different objectives impedes the agent to find a well-performing policy for the primal task, as shown by Fig. 1 which compares the policy gradients of three cases, namely the one without novel policy seeking, novelty seeking with multi-objective optimization and novelty seeking with constrained optimization methods, respectively.

In this work we take into consideration both the novelty of the learning policies and the performance of the primal task when addressing the problem of novel policy seeking. To this end we propose to seek novel policies under a formulation of constrained optimization. Two algorithms under such a formulation are designed to seek novel policies while keeping their performances of the primal task, avoiding excessive novelty seeking. As a result, the performances of our learned novel policies can be guaranteed and even further improved.

Our contributions can be summarized in three-folds. Firstly, we introduce a new metric to compute the difference between policies with instant feedback at every timestep; Secondly, we propose a constrained optimization formulation for novel policy seeking and design two practical algorithms resembling two approaches in constrained optimization literature; Thirdly, we evaluate our proposed algorithms on the MuJoCo locomotion environments, showing the advantages of these constrained optimization novelty-seeking methods which can generate a series of diverse and well-performing policies over previous multi-objective novelty seeking methods.

## 2 RELATED WORK

**Intrinsic motivation methods** In previous work, different approaches are proposed to provide intrinsic motivation or intrinsic reward as a supplementary to the primal task reward for better exploration (Houthooft et al., 2016; Pathak et al., 2017; Burda et al., 2018a;b; Liu et al., 2019). All those approaches leverage the weighted sum of two rewards, the primal rewards provided by environments, and intrinsic rewards that provided by different heuristics. On the other hand, the work of DIAYN and DADS (Eysenbach et al., 2018; Sharma et al., 2019) learn diverse skills without extrinsic reward. Those approaches focus on decomposing diverse skills of a single policy, while our work focuses on learning diverse behaviors among a batch of policies for the same task.

**Diverse policy seeking methods** The work of Such et al. shows that different RL algorithms may converge to different policies for the same task (Such et al., 2018). On the contrary, we are interested in how to learn different policies through a single learning algorithm with the capability of avoiding local optimum. The work of Pugh et al. establishes a standard framework for understanding and comparing different approaches to search for quality diversity (QD) (Pugh et al., 2016). Conti et al. proposes a solution which avoids local optima as well as achieves higher performance by adding novelty search and QD to evolution strategies (Conti et al., 2018). The Task-Novelty Bisector (TNB) (Zhang et al., 2019) aims to solve novelty seeking problem by jointly optimize the extrinsic rewards and novelty rewards defined by an auto-encoder. In this work, one of the two proposed methods is closely related to TNB, but is adapted to the constrained optimization formulation.

**Constrained Markov Decision Process** The Constrained Markov Decision Process (CMDP) (Altman, 1999) considers the situation where an agent interact with the environment under certain constraints. Formally, the CMDP can be defined as a tuple $(\mathcal{S}, \mathcal{A}, \gamma, r, c, C, P, s_0)$, where $\mathcal{S}$ and $\mathcal{A}$ are the state and action space; $\gamma \in [0, 1)$ is a discount factor; $r : \mathcal{S} \times \mathcal{A} \times \mathcal{S} \to \mathbb{R}$ and $c : \mathcal{S} \times \mathcal{A} \times \mathcal{S} \to \mathbb{R}$ denote the reward function and cost function; $C \in \mathbb{R}^+$ is the upper bound of permitted expected cumulative cost; $P(\cdot|s, a) : \mathcal{S} \times \mathcal{A} \to \mathcal{S}$ denotes the transition dynamics, and $s_0$ is the initial state. Denote the Markovian policy class as $\Pi$, where $\Pi = \{\pi : \mathcal{S} \times \mathcal{A} \to [0, 1], \sum_a \pi(a|\pi) = 1\}$ The learning objective of a policy for CMDP is to find a $\pi^* \in \Pi$, such that

$$\pi^* = \max_{\pi \in \Pi} \mathbb{E}_{\tau \sim \pi, s' \sim P}[\sum_{t=0}^{\infty} \gamma^t r(s, a, s')], \quad \text{s.t.} \quad \mathbb{E}_{\tau \sim \pi, s' \sim P}[\sum_{t=0}^{\infty} \gamma^t c(s, a, s')] \leq C, \quad (1)$$

where $\tau$ indicates a trajectory $(s_0, a_0, s_1, ...)$ and $\tau \sim \pi$ represents the distribution over trajectories following policy $\pi$: $a_t \sim \pi(\cdot|s_t), s_{t+1} \sim P(\cdot|s_t, a_t); t = 0, 1, 2, ...$. Previous literature provide several approaches to solve CMDP (Achiam et al., 2017; Chow et al., 2018; Ray et al., 2019), and in this work we include the CPO (Achiam et al., 2017) as baseline according to Ray et al. (2019).

## 3 METHODOLOGY

In Sec.3.1, we start with defining a metric space that measures the difference between policies, which is the fundamental element for the proposed methods. In Sec.3.2, we develop a practical estimation method for this metric. Sec.3.3 describes the formulation of constrained optimization on novel policy seeking. The implementations of two practical algorithms are further introduced in Sec.3.4.

We denote the policies as $\{\pi_{\theta_i}; \theta_i \in \Theta, i = 1, 2, ...\}$, wherein $\theta_i$ represents parameters of the $i$-th policy, $\Theta$ denotes the whole parameter space. In this work, we focus on improving the behavior diversity of policies from PPO (Schulman et al., 2017), thus we use $\Theta$ to represent $\Theta_{\text{PPO}}$ in this paper. It is worth noting that the proposed methods can be easily extended to other RL algorithms (Schulman et al., 2015; Lillicrap et al., 2015; Fujimoto et al., 2018; Haarnoja et al., 2018). To simplify the notation, we omit $\pi$ and denote a policy $\pi_{\theta_i}$ as $\theta_i$ unless stated otherwise.

### 3.1 MEASURING THE DIFFERENCE BETWEEN POLICIES

In this work, we use the Wasserstein metric $W_p$ (Rüschendorf, 1985; Villani, 2008; Arjovsky et al., 2017) to measure the distance between policies. Concretely, in this work we consider the Gaussian-parameterized policies, where the $W_p$ over two policies can be written in the closed form $W_2^2(\mathcal{N}(m_1, \Sigma_1), \mathcal{N}(m_2, \Sigma_2)) = ||m_1 - m_2||^2 + \text{tr}[\Sigma_1 + \Sigma_2 - 2(\Sigma_1^{1/2}\Sigma_2\Sigma_1^{1/2})^{1/2}]$ as $p = 2$, where $m_1, \Sigma_1, m_2, \Sigma_2$ are mean and covariance metrics of the two normal distributions. In the following of this paper, we use $D_W$ to denote the $W_2$ and it is worth noting that when the covariance matrix is identical, the trace term disappears and only the term involving the means remains, i.e., $D_W = |m_1 - m_2|$ for Dirac delta distributions located at points $m_1$ and $m_2$. This diversity metric satisfies the three properties of a metric, namely identity, symmetry as well as triangle inequality.

**Proposition 1** (Metric Space $(\Theta, \overline{D}_W^q)$). *The expectation of $D_W(\cdot, \cdot)$ of two policies over any state distribution $q(s)$:*

$$\overline{D}_W^q(\theta_i, \theta_j) := \mathbb{E}_{s \sim q(s)}[D_W(\theta_i(a|s), \theta_j(a|s))], \quad (2)$$

*is a metric on $\Theta$, thus $(\Theta, \overline{D}_W^q)$ is a metric space.*

The proof of Proposition 1 is straightforward. It is worth mentioning that Jensen Shannon divergence $D_{JS}$ or Total Variance Distance $D_{TV}$ (Endres & Schindelin, 2003; Fuglede & Topsoe, 2004; Schulman et al., 2015) can also be applied as alternative metric spaces, we choose $D_W$ in our work for that the Wasserstein metric better preserves the continuity (Arjovsky et al., 2017).

On top of the metric space $(\Theta, \overline{D}_W^q)$, we can then compute the novelty of a policy as follows.

**Definition 1** (Novelty of Policy). *Given a reference policy set $\Theta_{ref}$ such that $\Theta_{ref} = \{\theta_i^{ref}, i = 1, 2, ...\}, \Theta_{ref} \subset \Theta$, the novelty $\text{U}(\theta|\Theta_{ref})$ of policy $\theta$ is the minimal difference between $\theta$ and all policies in the reference policy set, i.e.,*

$$\text{U}(\theta|\Theta_{ref}) := \min_{\theta_j \in \Theta_{ref}} \overline{D}_W^q(\theta, \theta_j). \quad (3)$$

Consequently, to encourage the discovery of novel policies discovery, typical novelty-seeking methods tend to directly maximize the novelty of a new policy, i.e., $\max_\theta U(\theta|\Theta_{ref})$, where the $\Theta_{ref}$ includes all existing policies.

## 3.2 Estimation of $\overline{D}_W^q(\theta_i, \theta_j)$ and the Selection of $q(s)$

In practice, the calculation of $\overline{D}_W^q(\theta_i, \theta_j)$ is based on Monte Carlo estimation where we need to sample $s$ from $q(s)$. Although in Eq.(2) $q(s)$ can be selected simply as a uniform distribution over the state space, there remains two obstacles: first, in a finite state space we can get precise estimation after establishing ergodicity, but problem arises when facing continuous state spaces due to the difficulty of efficiently obtaining enough samples; second, when $s$ is sampled from a uniform distribution $q$, we can only get *sparse* episodic reward instead of *dense* online reward which is more useful in learning. Therefore, we make an approximation here based on importance sampling.

Formally, we denote the domain of $q(s)$ as $\mathcal{S}_q \subset \mathcal{S}$ and assume $q(s)$ to be a uniform distribution over $\mathcal{S}_q$, without loss of generality in later analysis. Notice $\mathcal{S}_q$ is closely related to the algorithm being used in generating trajectories (Henderson et al., 2018). As we only care about the reachable regions of a certain algorithm (in this work, PPO), the domain $\mathcal{S}_q$ can be decomposed by $\mathcal{S}_q = \lim_{N \to \infty} \bigcup_{i=1}^N \mathcal{S}_{\theta_i}$, where $\mathcal{S}_{\theta_i}$ denotes all the possible states a policy $\theta_i$ can visit given a starting state distribution.

In order to get online-reward, we estimate Eq.(2) with

$$\overline{D}_W^q(\theta_i, \theta_j) = \mathbb{E}_{s \sim q(s)}[D_W(\theta_i(a|s), \theta_j(a|s))] = \mathbb{E}_{s \sim \rho_{\theta_i}(s)}\left[\frac{q(s)}{\rho_{\theta_i}(s)}D_W(\theta_i(a|s), \theta_j(a|s))\right], \quad (4)$$

where we use $\rho_\theta(s)$ to denote the stationary state visitation frequency under policy $\theta$, i.e., $\rho_\theta(s) = P(s_0 = s|\theta) + P(s_1 = s|\theta) + ... + P(s_T = s|\theta)$ in finite horizon problems. We propose to use the averaged stationary visitation frequency as $q(s)$, e.g., for PPO, $q(s) = \overline{\rho}(s) = \mathbb{E}_{\theta \sim \Theta_{PPO}}[\rho_\theta(s)]$. Clearly, choosing $q(s) = \overline{\rho}(s)$ will be much better than choosing a uniform distribution as the importance weight will be closer to 1. Such an importance sampling process requires a necessary condition that $\rho_{\theta_i}(s)$ and $q(s)$ have the same domain, which can be guaranteed by applying a sufficient exploration noise on $\theta$.

Another difficulty lies in the estimation of $\overline{\rho}(s)$, which is always intractable given a limited number of trajectories. However, during training, $\theta_i$ is a policy to be optimized and $\theta_j \in \Theta_{ref}$ is a fixed reference policy. The error introduced by approximating the importance weight as 1 will get larger when $\theta_i$ becomes more distinct from normal policies, at least in terms of the state visitation frequency. We may just regard increasing of the approximation error as the discovery of novel policies.

**Proposition 2** (Unbiased Single Trajectory Estimation). *The estimation of $\rho_\theta(s)$ using a single trajectory $\tau$ is unbiased.*

The Proposition 2 follows the usual trick in RL that uses a single trajectory to estimate the stationary state visitation frequency. Given the definition of novelty and a practically unbiased sampling method, the next step is to develop an efficient learning algorithm.

## 3.3 Constrained Optimization Formulation for Novel Policy Seeking

In the traditional RL paradigm, maximizing the expectation of cumulative rewards is commonly used as the objective. i.e., $\max_{\theta \in \Theta} \mathbb{E}_{\tau \sim \theta}[g]$, where $g = \sum_{t=0} \gamma^t r_t$ and $\tau \sim \theta$ denotes a trajectory $\tau$ sampled from the policy $\theta$.

To improve the diversity of different agents' behaviors, the learning objective must take both the reward from the primal task and the policy novelty into consideration. Previous approaches (Houthooft et al., 2016; Pathak et al., 2017; Burda et al., 2018a;b; Liu et al., 2019) often directly use the weighted sum of these two terms as the objective:

$$\max_{\theta \in \Theta} \mathbb{E}_{\tau \sim \theta}[g_{total}] = \max_{\theta \in \Theta} \mathbb{E}_{\tau \sim \theta}[\alpha \cdot g_{task} + (1 - \alpha) \cdot g_{int}], \quad (5)$$

where $0 < \alpha < 1$ is a weight hyper-parameter, $g_{task}$ is the reward from the primary task, and $g_{int} = \sum_{t=0} \gamma^t r_{int,t}$ is the cumulative intrinsic reward of the *intrinsic reward* $r_{int,t}$. In our case, the intrinsic reward is the novelty reward $r_{int} = \min_{\theta_j \in \Theta_{ref}} \overline{D}_W^{\overline{\rho}}(\theta, \theta_j)$. These methods can be

summarized as Weighted Sum Reward (WSR) methods (Zhang et al., 2019). Such an objective is sensitive to the selection of $\alpha$ as well as the formulation of $r_{\text{int}}$. For example, in our case formulating the novelty reward $r_{\text{int}}$ as $\min_{\theta_j} \overline{D}_W^{\rho}(\theta, \theta_j)$, $\exp\left[\min_{\theta_j} \overline{D}_W^{\rho}(\theta, \theta_j)\right]$ and $-\exp\left[-\min_{\theta_j} \overline{D}_W^{\rho}(\theta, \theta_j)\right]$ will lead to significantly different results as they determine the trade-offs in the two terms given $\alpha$. Besides, dilemma also arises in the selection of $\alpha$: while a large $\alpha$ may undermine the contribution of intrinsic reward, a small $\alpha$ could ignore the importance of the primal task, leading to the failure of an agent in solving the task.

To tackle such an issue, the crux is to deal with the conflict between different objectives. The work of Zhang et al. proposes the TNB, where the task reward is regarded as the dominant one while the novelty reward is regarded as subordinate Zhang et al. (2019). However, as TNB considers the novelty gradient all the time, it may hinder the learning process, e.g., Intuitively, well-performing policies should be more similar to each other than to random initialized policies. As a new random initialized policy is different enough from previous policies, considering the novelty gradient at beginning of training will result in a much slower learning process.

In order to tackle the above problems and adjust the extent of novelty in new policies, we propose to solve the novelty-seeking problem under the perspective of constrained optimization. The basic idea is as follows: while the task reward is considered as a learning objective, the novelty reward should be considered as a bonus instead of another objective, and should not impede the learning of the primal task. Fig. 1 illustrates how novelty gradients impede the learning of a policy: at the beginning of learning, a random initialized policy should in total learn to be more similar to a well-performing policy rather than be different. The seeking of novelty should not be taken into consideration all the time during learning. With such an insight, we change the multi-objective optimization problem in Eq.(5) into a constrained optimization problem as:

$$\max_{\theta \in \Theta} \quad f(\theta) = \mathbb{E}_{\tau \sim \theta}[g_{\text{task}}], \quad \text{s.t.} \quad g_t(\theta) = \bar{r}_{\text{int},t} - r_0 \geq 0, \forall t = 1, 2, ..., T, \tag{6}$$

where $r_0$ is a threshold indicating minimal permitted novelty, and $\bar{r}_{\text{int},t}$ denotes a moving average of $r_{\text{int},t}$. as we need not force every single action of a new agent to be different from others. Instead, we care more about the long-term differences. Therefore, we use cumulative novelty terms as constraints. Moreover, the constraints can be flexibly applied after the first $t_S$ timesteps (e.g., $t_S = 20$) for the consideration of similar starting sequences, so that the constraints can be written as $g_t(\theta) \geq 0, \forall t = t_S, ..., T$.

### 3.4 Practical Novel Policy Seeking Methods

We note here, WSR and TNB proposed in previous work (Zhang et al., 2019) can correspond to different approaches in constrained optimization problems, yet some important ingredients are missing. We improve TNB according to the Feasible Direction Method in constrained optimization and then propose the Interior Policy Differentiation (IPD) method according to the Interior Point Method in constrained optimization.

**WSR: Penalty Method**  The Penalty Method considers the constraints of Eq.(6) by putting constraint $g(\theta)$ into a penalty term, followed by solving the unconstrained problem

$$\max_{\theta \in \Theta} \quad f(\theta) + \frac{1 - \alpha}{\alpha} \min\{g(\theta), 0\}, \tag{7}$$

in an iterative manner. The limit of the above unconstrained problem when $\alpha \to 0$ then leads to the solution of the original constrained problem. As an approximation, WSR chooses a fixed weight $\alpha$, and uses the gradient of $\nabla_\theta f + \frac{1-\alpha}{\alpha} \nabla_\theta g$ instead of $\nabla_\theta f + \frac{1-\alpha}{\alpha} \nabla_\theta \min\{g(\theta), 0\}$, thus the final solution will intensely rely on the selection of $\alpha$.

**TNB: Feasible Direction Method**  The Feasible Direction Method (FDM) (Ruszczyński, 1980; Herskovits, 1998) solves the constrained optimization problem by finding a direction $\vec{p}$ where taking gradient upon will lead to increment of the objective function as well as constraints satisfaction, i.e., $\nabla_\theta f^{\mathrm{T}} \cdot \vec{p} > 0$, if $g > 0$ and $\nabla_\theta g^{\mathrm{T}} \cdot \vec{p} > 0$ otherwise. The TNB proposes to use a revised bisector of gradients $\nabla_\theta f$ and $\nabla_\theta g$ as $\vec{p}$,

$$\vec{p} = \begin{cases} \nabla_\theta f + \frac{|\nabla_\theta f|}{|\nabla_\theta g|} \nabla_\theta g \cdot \cos(\nabla_\theta f, \nabla_\theta g) & \text{if } \cos(\nabla_\theta f, \nabla_\theta g) \leq 0 \\ \nabla_\theta f + \frac{|\nabla_\theta f|}{|\nabla_\theta g|} \nabla_\theta g & \text{if } \cos(\nabla_\theta f, \nabla_\theta g) > 0 \end{cases} \tag{8}$$

Clearly, Eq.(8) satisfies the constraints but it is more strict than it as the $\nabla_\theta g$ term always exists during the optimization of TNB. Based on TNB, we provide a revised approach, named Constrained Task Novel Bisector (CTNB), which resembles better with FDM. Specifically, when $g > 0$, CTNB will not apply $\nabla_\theta g$ on $g$. It is clear that TNB is a special case of CTNB when the novelty threshold $r_0$ is set to infinity. We note that in both TNB and CTNB, the learning stride is fixed to be $\frac{|\nabla_\theta f| + |\nabla_\theta g|}{2}$ and may lead to problem when $\nabla_\theta f \to 0$, where the final optimization result will rely heavily on the selection of $g$, i.e., the shape of $g$ is crucial for the success of this approach.

**IPD: Interior Point Method**   The Interior Point Method (Potra & Wright, 2000; Dantzig & Thapa, 2006) is another approach used to solve the constrained optimization problem. Thus here we solve Eq.(6) using the Interior Policy Differentiation (IPD), which can be regarded as an analogy of the Interior Point Method. In the vanilla Interior Point Method, the constrained optimization problem in Eq.(6) is solved by reforming it to an unconstrained form with an additional barrier term $-\alpha \log g(\theta)$ in the objective as $\max_{\theta \in \Theta} \quad f(\theta) - \alpha \log g(\theta)$, or more precisely in our problem with the formulation with Eq.(6) we have $\max_{\theta \in \Theta} \quad \mathbb{E}_{\tau \sim \theta}[g_{\text{task}} - \sum_{t=0}^{T} \alpha \log(\bar{r}_{\text{int},t} - r_0)]$, where $\alpha > 0$ is the barrier factor. Besides the log barrier term, there are other choices like $\alpha \frac{1}{g(\theta)}$ can be used and the objective becomes $\max_{\theta \in \Theta} \quad f(\theta) + \alpha \frac{1}{g(\theta)}$. As $\alpha$ is small, the barrier term will introduce only minuscule influence on the objective. On the other hand, when $\theta$ get closer to the barrier, the objective will increase rapidly. The limits when $\alpha \to 0$ then lead to the solution of Eq.(6). The convergence of such methods are provided in previous works Conn et al. (1997); Wright (2001).

However, directly applying IPM is computationally expensive and numerically unstable. In this work, we propose a simple yet novel heuristic method that resembles the idea of barrier methods: we implicitly apply such barrier terms by providing termination signals in interactions with the environments. Our method can be regarded as revising the primal task MDP into a new one in which the behaviors of agents must satisfy novelty constraints. Specifically, in the RL paradigm, the learning procedure of an agent is determined by the experiences collected during interactions with the environment and the sampling strategy used to filter experiences in the calculation of policy gradients. Since the learning process is based on sampled transitions, a more natural way can thus be used to perform the constrained optimization. We can simply bound the collected transitions in the feasible region by permitting previously trained $M$ policies $\theta_i \in \Theta_{\text{ref}}, i = 1, 2, ..., M$ sending termination signals during the training process of new agents. In other words, we implicitly bound the feasible region by terminating any new agent that steps outside it.

Consequently, during the training process, all valid samples we collected are inside the feasible region, which means these samples are less likely to appear in previously trained policies. At the end of the training, we obtain a new policy that has sufficient novelty. In this way, we no longer need to consider the trade-off between intrinsic and extrinsic rewards deliberately. The learning process of IPD is thus more robust and no longer suffers from the objective inconsistency.

## 4 EXPERIMENTS

According to Proposition 2, the novelty reward $r_{int}$ in Eq.(6) under our novelty metric can be unbiasedly approximated by $r_{\text{int}} = \min_{\theta_j \in \Theta_{ref}} \overline{D}_W^{\rho_\theta}(\theta(a|s_t), \theta_j(a_j|s_t))$. We thus utilize this novelty metric directly throughout our experiments. We apply different novel policy seeking methods, namely WSR, TNB, CTNB, and IPD, to the backbone RL algorithm PPO (Schulman et al., 2017). The extension to other popular RL algorithms is straightforward. More implementation details are depicted in Appendix D. Experiments in the work of Henderson et al. show that one can simply change the random seeds before training to get policies that perform differently Henderson et al. (2018). Therefore, we also use PPO with varying random seeds as a baseline method for novel policy seeking. And we use the averaged differences between policies learned by this baseline as the default threshold in CTNB and IPD. Algorithm 1 and Algorithm 2 show the pseudo code of IPD and CTNB based on PPO, where the blue lines show the addition to the primal PPO algorithm.

### 4.1 THE MUJOCO ENVIRONMENT

We evaluate our proposed method on the OpenAI Gym based on the MuJoCo engine (Brockman et al., 2016; Todorov et al., 2012). Concretely, we test on three locomotion environments, the Hopper-v3

**Algorithm 1** IPD

**Input:**
  (1) a behavior policy $\theta_{old}$;
  (2) a set of previous policies $\{\theta_j\}, j = 1, 2, ..., M$;
  (3) a novelty metric $U(\theta, \{\theta_j\}|\rho) = U(\theta, \{\theta_j\}|\tau) = \min_{\theta_j} \overline{D}_W^\tau(\theta, \theta_j)$;
  (4) a novelty threshold $r_0$ and starting point $t_S$
Initialize $\theta_{old}$;
  **for** *iteration* $= 1, 2, ...$ **do**
    **for** $t = 1, 2, ..., T$ **do**
      Step the environment by taking action $a_t \sim \theta_{old}$ and collect transitions;
      **if** $U(\theta_{old}, \{\theta_j\}|\tau) - r_0 < 0$ *AND* $t > t_S$ **then**
        | Break this episode;
      **end**
    **end**
    Update policy parameters based on sampled data;
**end**

**Algorithm 2** Constrained TNB

**Input:**
  (1) to (4) same as Algo.1
  (5) a value network for cost $V_c$
Initialize $\theta_{old}$;
  **for** *iteration* $= 1, 2, ...$ **do**
    **for** $t = 1, 2, ..., T$ **do**
      Step the environment by taking action $a_t \sim \theta_{old}$ and collect transitions;
    **end**
    Compute advantage of reward $\hat{A}_{r,1}, ..., \hat{A}_{r,T}$
    Compute advantage of cost $\hat{A}_{c,1}, ..., \hat{A}_{c,T}$
    Optimize surrogate loss related to reward $\mathcal{L}_r^{\text{CLIP}}$ in PPO w.r.t. $\theta$, with gradient $g_r = \nabla_\theta \mathcal{L}_r^{\text{CLIP}}$
    Optimize surrogate loss related to cost $\mathcal{L}_c^{\text{CLIP}}$ in PPO w.r.t. $\theta$, with gradient $g_c = -\nabla_\theta \mathcal{L}_c^{\text{CLIP}}$
    **if** $U(\theta_{old}, \{\theta_j\}|\tau) - r_0 < 0$ **then**
      | Calculate $\vec{p}$ according to Eq.(8) with $g_r$ and $g_c$
    **else**
      | Calculate $\vec{p}$ with $g_r$
    **end**
    Update policy parameters
**end**

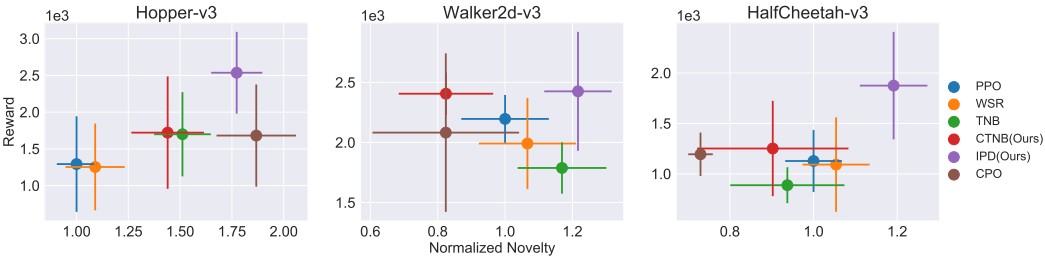

Figure 2: The performance and novelty comparison of different methods in Hopper-v3, Walker2d-v3 and HalfCheetah-v3 environments. The value of novelty is normalized to relative novelty by regarding the averaged novelty of PPO policies as the baseline. The results are from 10 policies of each method, with the points show their mean and lines show their standard deviation.

(11 observations and 3 actions), Walker2d-v3 (11 observations and 6 actions), and HalfCheetah-v3 (17 observations and 6 actions). Although relaxing the healthy termination thresholds in Hopper and Walker may permit more visible behavior diversity, all the environment parameters are set as default values in our experiments to demonstrate the generality of our method.

### 4.1.1 COMPARISON ON NOVELTY AND PERFORMANCE

We implement WSR, TNB, CTNB, and IPD using the same hyper-parameter settings per environment. And we also apply CPO Achiam et al. (2017) as a baseline as a solution of CMDP. For each method, we first train 10 policies using PPO with different random seeds. Those PPO policies are used as the primal reference policies, and then we train 10 novel policies that try to be different from previous reference policies. Concretely, in each method, the $1st$ novel policy is trained to be different from the previous 10 PPO policies, and the $2nd$ should be different from the previous 11 policies, and so on. More implementation details are depicted in Appendix D.

Table 1: The Reward and Success Rate of 10 Policies. Our CTNB and IPD beat CPO, TNB and WSR in all three environments. Constrained optimization approaches outperforms multi-objective methods. Results are generated from 5 random seeds.

| | Reward | | | Success Rate | | |
|---|---|---|---|---|---|---|
| Environment | Hopper | Walker2d | HalfCheetah | Hopper | Walker2d | HalfCheetah |
| PPO | $1292 \pm 650$ | $2196 \pm 200$ | $1127 \pm 308$ | 0.5 | 0.5 | 0.5 |
| WSR | $1253 \pm 591$ | $1992 \pm 380$ | $1091 \pm 469$ | 0.6 | 0.3 | 0.3 |
| TNB | $1699 \pm 573$ | $1788 \pm 214$ | $887 \pm 178$ | 0.8 | 0.0 | 0.1 |
| CPO | $1681 \pm 696$ | $2082 \pm 660$ | $1194 \pm 215$ | 0.8 | 0.6 | 0.8 |
| CTNB (Ours) | $1721 \pm 765$ | $\mathbf{2405 \pm 177}$ | $1251 \pm 473$ | 0.8 | **0.9** | 0.5 |
| IPD (Ours) | $\mathbf{2536 \pm 557}$ | $2282 \pm 206$ | $\mathbf{1875 \pm 533}$ | **1.0** | 0.6 | **0.9** |

Fig. 2 shows our experimental results in terms of novelty (the x-axis) and the performance (the y-axis). Policies close to the upper right corner are the more novel ones with higher performance. In all environments, the performance of CTNB, IPD and CPO outperforms WSR and TNB, showing the advantage of constrained optimization approaches in novel policy seeking. Specifically, the results of CTNB are all better than their multi-objective counterparts, i.e., the results from TNB, showing the superiority of seeking novel policies with constrained optimization. Moreover, the IPD method provides more novelty than CTNB and CPO, while the primal task performances are still guaranteed.

Comparisons of the task-related rewards are carried out in Table 1, where among all the four methods, IPD provides sufficient diversity with minimum loss of performance. Instead of performance decay, we find IPD is able to find better policies in the environment of Hopper and HalfCheetah. Moreover, in the Hopper environment, while the agents trained with PPO tend to fall into the same local minimum. (e.g., they all jump as far as possible and then terminate this episode. On the contrary, PPO with IPD keeps new agents away from falling into the same local minimum, because once an agent has reached some local minimum, agents learned later will try to avoid this region due to the novelty constraints. Such property shows that IPD can enhance the traditional RL schemes to tackle the local exploration challenge (Tessler et al., 2019; Ciosek et al., 2019). A similar feature brings about reward growth in the environment of HalfCheetah. Detailed analysis and discussions are developed in Appendix E.

### 4.1.2 SUCCESS RATE OF EACH METHOD

In addition to averaged reward, we also use the success rate as another metric to compare the performance of different approaches. Roughly speaking, the success rate evaluates the stability of each method in terms of generating a policy that performs as good as the policies PPO generates. In this work, we regard a policy successful when its performance achieves at least as good as the median performance of policies trained with PPO. To be specific, we use the median of the final performance of PPO as the baseline, and if a novel policy, which aims at performing differently to solve the same task, surpasses the baseline during its training process, it will be regarded as a successful policy. By definition, the success rate of PPO is 0.5 as a baseline for every environment. Table 1 shows the success rate of all the methods. The results show that all constrained novelty seeking methods (CTNB, IPD, CPO) can surpass the average baseline during training, while the multi-objective optimization approaches normally can not. Thus the performance of constrained novelty seeking methods can always be insured.

## 5 CONCLUSION

In this work, we rethink the novel policy seeking problem under the perspective of constrained optimization. We first introduce a new metric to measure the distances between policies, and then give a definition of policy novelty. Based on the formulation of constrained optimization, we come up with two practical algorithms for novel policy seeking, namely the Constrained Task Novel Bisector (CTNB), and the Interior Policy Differentiation (IPD). Our experimental results demonstrate that the proposed method can effectively learn various well-performing yet diverse policies, outperforming previous methods which are under the multi-objective formulation.

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

APPENDIX

## A  METRIC SPACE

**Definition 2.** *A metric space is an ordered pair $(M, d)$ where $M$ is a set and $d$ is a metric on $M$, i.e., a function $d \colon M \times M \to \mathbb{R}$ such that for any $x, y, z \in M$, the following holds:*
1.  *$d(x, y) \geq 0, d(x, y) = 0 \Leftrightarrow x = y$,*
2.  *$d(x, y) = d(y, x)$,*
3.  *$d(x, z) \leq d(x, y) + d(y, z)$.*

## B  PROOF OF PROPOSITION 1

The first two properties are obviously guaranteed by $\overline{D}_W^\rho$. As for the triangle inequality,

$$\mathbb{E}_{s \sim \rho(s)}[D_W(\theta_i(s), \theta_k(s)]$$
$$= \mathbb{E}_{s \sim \rho(s)}[\sum_{l=1}^{|\mathcal{A}|} |\theta_i(s) - \theta_k(s)|]$$
$$= \mathbb{E}_{s \sim \rho(s)}[\sum_{l=1}^{|\mathcal{A}|} |\theta_i(s) - \theta_j(s) + \theta_j(s) - \theta_k(s)|]$$
$$\leq \mathbb{E}_{s \sim \rho(s)}[\sum_{l=1}^{(|\mathcal{A}|} |\theta_i(s) - \theta_j(s)| + |\theta_j(s) - \theta_k(s)|)]$$
$$= \mathbb{E}_{s \sim \rho(s)}[\sum_{l=1}^{|\mathcal{A}|} |\theta_i(s) - \theta_j(s)|] + \mathbb{E}_{s \sim \rho(s)}[\sum_{l=1}^{|\mathcal{A}|} |\theta_j(s) - \theta_k(s)|]$$
$$= \mathbb{E}_{s \sim \rho(s)}[D_W(\theta_i(s), \theta_j(s)] + \mathbb{E}_{s \sim \rho(s)}[D_W(\theta_j(s), \theta_k(s)]$$

## C  PROOF OF PROPOSITION 2

$$\rho_\theta(s) = P(s_0 = s|\theta) + P(s_1 = s|\theta) + ... + P(s_T = s|\theta)$$
$$\overset{L.L.N.}{=} \lim_{N \to \infty} \frac{\sum_{i=1}^N I(s_0 = s|\tau_i)}{N} + \frac{\sum_{i=1}^N I(s_1 = s|\tau_i)}{N} + ... + \frac{\sum_{i=1}^N I(s_T = s|\tau_i)}{N}$$
$$= \lim_{N \to \infty} \frac{\sum_{j=0}^T \sum_{i=1}^N I(s_j = s|\tau_i)}{N}$$
$$\overline{\rho}_\theta(s) = \sum_{i=1}^N \sum_{j=0}^T \frac{I(s_j = s|\tau_i)}{N}$$
$$\mathbb{E}[\overline{\rho}_\theta(s) - \rho_\theta(s)] = 0$$

## D  IMPLEMENTATION DETAILS

**Calculation of $D_W$**  We use deterministic part of policies in the calculation of $D_W$, i.e., we remove the Gaussian noise on the action space in PPO and use $D_W(a_1, a_2) = |a_1 - a_2|$.

**Network Structure**  We use MLP with 2 hidden layers as our actor models in PPO. The first hidden layer is fixed to have 32 units. We choose to use 10, 64 and 256 hidden units for the three tasks respectively in all of the main experiments, after taking the success rate, performance and computation expense (i.e. the preference to use less unit when the other two factors are similar) into consideration.

Figure 3: The performance under different novelty thresholds in the Hopper, Walker and HalfCheetah environments. The results are collected from 10 learned policies based on PPO. The box extends from the lower to upper quartile values of the data, with a line at the median. The whiskers extend from the box to show the range of the data. Flier points are those past the end of the whiskers.

**Training Timesteps**   We fix the training timesteps in our experiments. The timesteps are fixed to be 1M in Hopper-v3, 1.6M for Walker2d-v3 and 3M for HalfCheetah-v3.

## E   DISCUSSION

### E.1   NOVEL POLICY SEEKING WITHOUT PERFORMANCE DECAY

Multi-objective formulation of novel policy seeking has the risk of sacrificing the primal performance as the overall objective needs to consider both novelty and primal task rewards. On the contrary, under the perspective of constrained optimization, there will be no more trade-off between novelty and final reward as the only objective is the task reward. Given a certain novelty threshold, the algorithms tend to find the optimal solution in terms of task reward under constraints, thus the learning process becomes more controllable and reliable, i.e., one can utilize the novelty threshold to control the degree of novelty.

Intuitively, the proper magnitude of the novelty threshold will lead to more exploration among a population of policies, thus the performance of latter found policies may be better than or at least as good as those trained without novelty seeking. However, when a larger magnitude of novelty threshold is applied, the performance of found novel policies will decrease because finding a feasible solution will get harder under more strict constraints. Fig. 3 shows our ablation study on adjusting the thresholds, which verifies our intuition.

### E.2   CURRICULUM LEARNING IN HALFCHEETAH

Moreover, we observe a kind of auto-curriculum learning behavior in the learning of HalfCheetah, which may also help to understand the performance improvement in this environment. The environment of HalfCheetah is different from the other two in that there is no explicit early termination signal in its default setting (i.e., there is no explicit threshold for the states, exceeds which would trigger a termination). At the beginning of the learning, a PPO agent always acts randomly and keep twitching without moving, resulting in massive repeated and trivial samples and large control costs. Contrarily, in the learning of IPD, the agent can receive termination signals since repeated behaviors break the novelty constraint, preventing it from wasting too much effort acting randomly. Moreover, such termination signals also encourage the agent to imitate previous policies to get out of random explorations at the starting stage, avoiding heavy control costs while receiving less negative rewards. After that, the agent begins to learn to behave differently to pursue higher positive rewards. From this point of view, the learning process can be interpreted as a kind of implicit curriculum, which saves lots of interactions with the environment, improves the sample efficiency and therefore achieves better performance in the given learning timesteps.

## F   ADDITIONAL EXPERIMENTS

We run additional experiments on the Ant-v3 environment to show the performance of our algorithm on the more complicated continuous control task (with 111-dim state space and 8-dim action space).

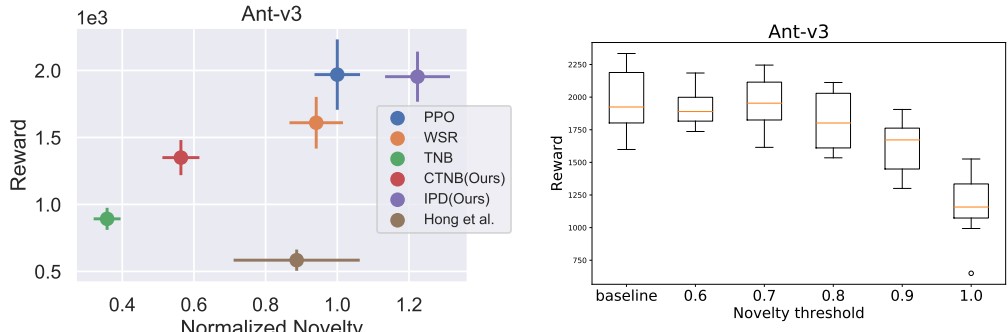

Figure 4: (Left): performance comparison in terms of novelty and task performance of PPO, WSR and IPD in the Ant-v3 environment. We also include the work of (Hong et al., 2018) as a comparison. (Right): performance under different choices on the novelty threshold in the Ant-v3 environment. The results are collected from 10 learned policies based on PPO. The box extends from the lower to upper quartile values of the data, with a line at the median. The whiskers extend from the box to show the range of the data. Flier points are those past the end of the whiskers.

Results are shown in Figure 4. Our method of IPD achieves on-par performance with PPO but improves the novelty between policies by 20%.

Notably, both the performance and novelty of CTNB in Ant-v3 is better than its multi-objective optimization counterpart, the TNB, we attribute the reason to the limited training timesteps in TNB and CTNB: in limited training timesteps (3M timesteps in Ant-v3), the policies trained with CTNB and TNB can not converge to well-performing policies, and therefore the behavior difference between those policies are limited (even less than PPO). On the contrary, the method of IPD does not fuse the gradient of primal task reward and the novelty reward, thus similar learning efficiency can be achieved and result in well-performing policies.

This experiment demonstrates our claim on the superiority of constrained optimization perspectives of novelty-seeking again: too much pursuance of the novelty will hinder the primal task performance as well as hinder the generating of both diverse and well-performing policies.

## G  VISUALIZE DIVERSITY

### G.1  THE FOUR REWARD MAZE PROBLEM

We first utilize a basic 2-D environment named Four Reward Maze as a diagnostic environment where we can visualize learned policies directly. In this environment, four positive rewards of different values (e.g., $+5, +5, +10, +1$ for top, down, left and right respectively) are assigned to four middle points with radius 1 on each edge in a 2-D $N \times N$ square map. We use $N = 16$ in our experiments. The observation of a policy is the current position and the agent will receive a negative reward of $-0.01$ at each timestep except stepping into the reward regions. Each episode starts from a randomly initialized position and the action space is limited to $[-1, 1]$. The performance of each agent is evaluated by the averaged performances over 100 trials.

Results are shown in Fig. 5, where the behaviors of the PPO agents are quite similar, suggesting the diversity provided by random seeds is limited. WSR and TNB solve the novelty-seeking problem from the multi-objective optimization formulation, they thus suffer from the unbalance between performance and novelty. While WSR and TNB both provide sufficient novelty, performances of agents learned by WSR decay significantly, so did TNB due to an encumbered learning process, as we analyzed in Sec.3.3. Both CTNB and IPD, solving the task with novelty-seeking from the constrained optimization formulation, provide evident behavior diversity and perform recognizably better than TNB and WSR.

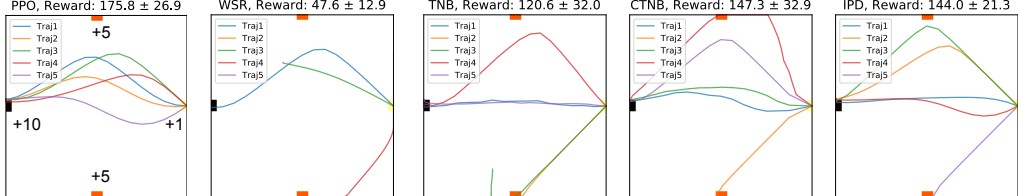

Figure 5: Experimental results on the Four Reward Maze Problem. We generate 5 policies with different novelty seeking methods, and use the PPO with different random seeds as baseline. In each figure, the 5 lines indicate 5 trajectories when the game is started from the right hand side. It worth noting that the results of WSR, CTNB and IPD are associated with the parameters of weights or threshold. We set the weight parameter in WSR as $10$ to make the two reward terms comparable, and set the thresholds in CTNB and IPD as the averaged novelty between policies trained with PPO. All policies are trained with $6.1 \times 10^3$ episodes.

## G.2 MUJOCO LOCOMOTION

In this section, we provide some qualitative results of IPD on the Mujoco locomotion tasks. In all of our experiments we use the vanilla Mujoco locomotion benchmarks, with the default settings on defining healthy states. Although otherwise the visualization of learned policies might become more diverse (e.g., a Hopper agent may learn to stand-up after falling down while another agent may learn to move forward on the ground if we set the $z$-axis healthy threshold as $0$).

With the method of IPD, the Hopper policies (Figure 6) learns to jump further and avoids falling down rather instead of just jumping and falling down (Figure 7). In the Walker2d environment, the color of purple indicates the left leg is visible. It can be seen that the IPD policies (Figure 8) learn to use both left and right legs in walking, while the PPO policies usually learn jumping. (Figure 9). In HalfCheetah, the IPD policies (Figure 10) perform much better than the PPO policies (Figure 11). The IPD policies leran to run with head-downward (Figure 10 line 1), head-upward (Figure 10 line 3), and forward (Figure 10 line 5) while the PPO policies are always head-downward.

In Hopper and HalfCheetah, IPD is able to improve the primal task performance by avoiding always getting trapped in some certain sub-optimal behaviors.

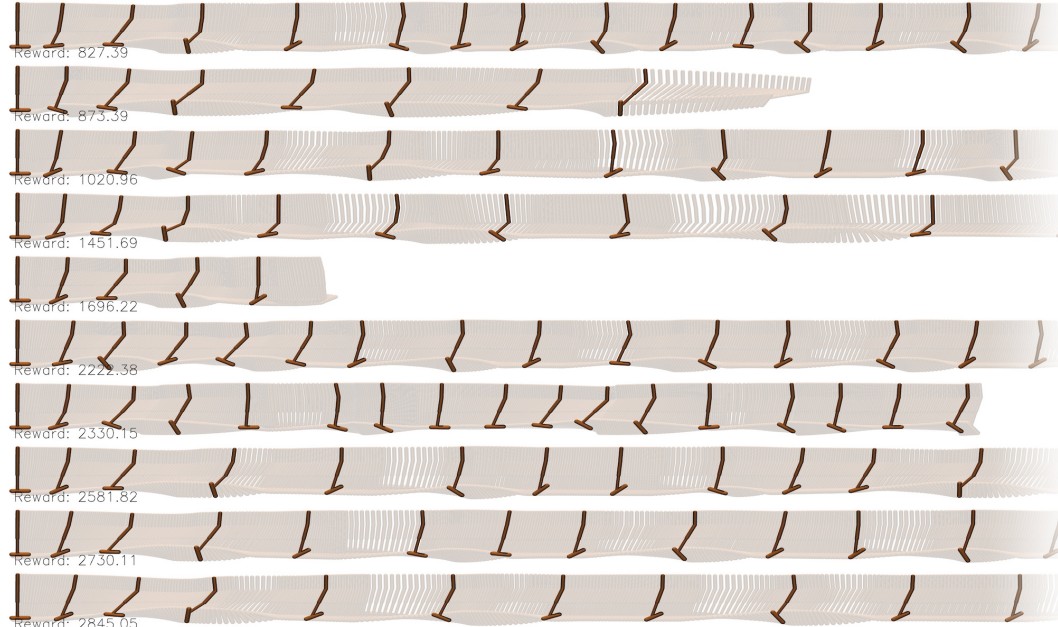

Figure 6: The visualization of policy behaviors of agents trained by our method in Hopper-v3 environment. Agents learn to jump with different strides.

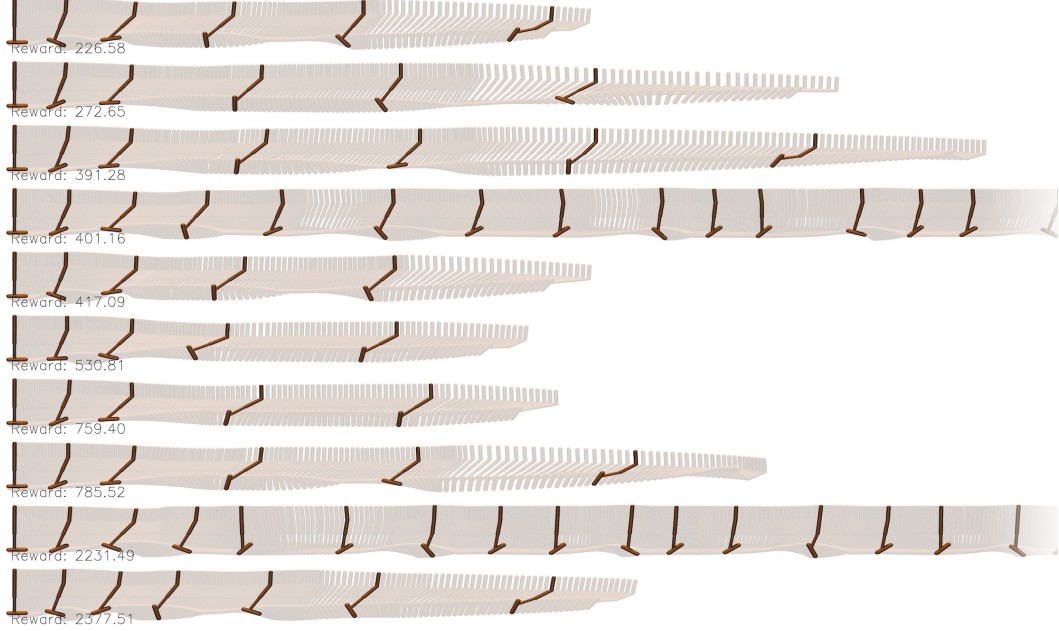

Figure 7: The visualization of policy behaviors of agents trained by PPO in Hopper-v3 environment. Most agents learn a policy that can be described as *Jump as far as possible and fall down*, leading to relative poor performance.

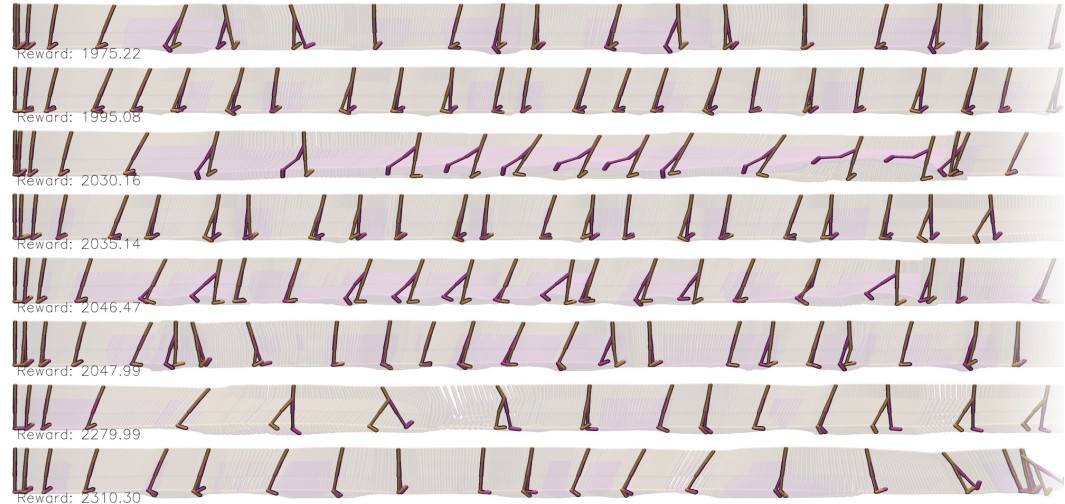

Figure 8: The visualization of policy behaviors of agents trained by our method in Walker2d-v3 environment. Instead of bouncing at the ground using both legs, our agents learns to use both legs to step forward.

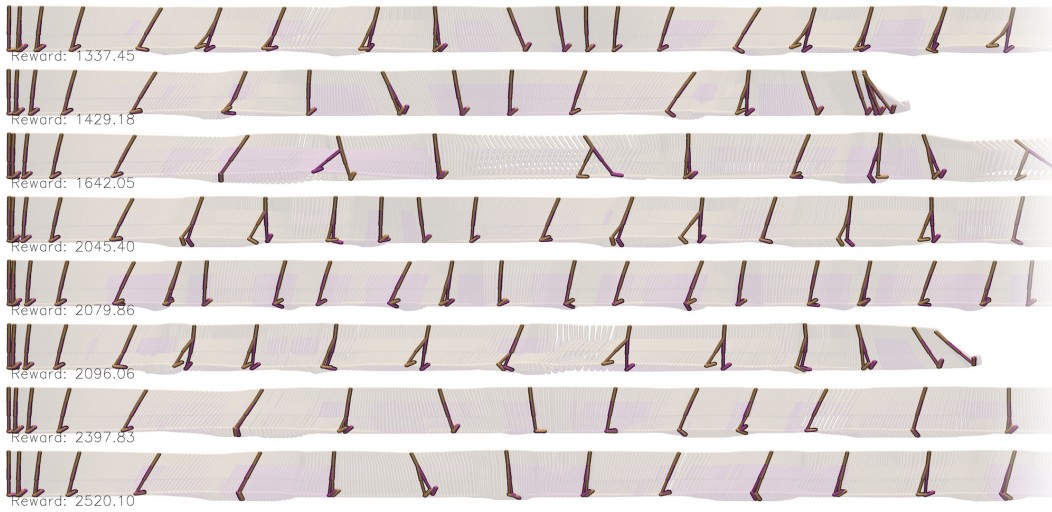

Figure 9: The visualization of policy behaviors of agents trained by PPO in Walker2d-v3 environment. Most of the PPO agents only learn to use their right leg to support the body and jump forward.

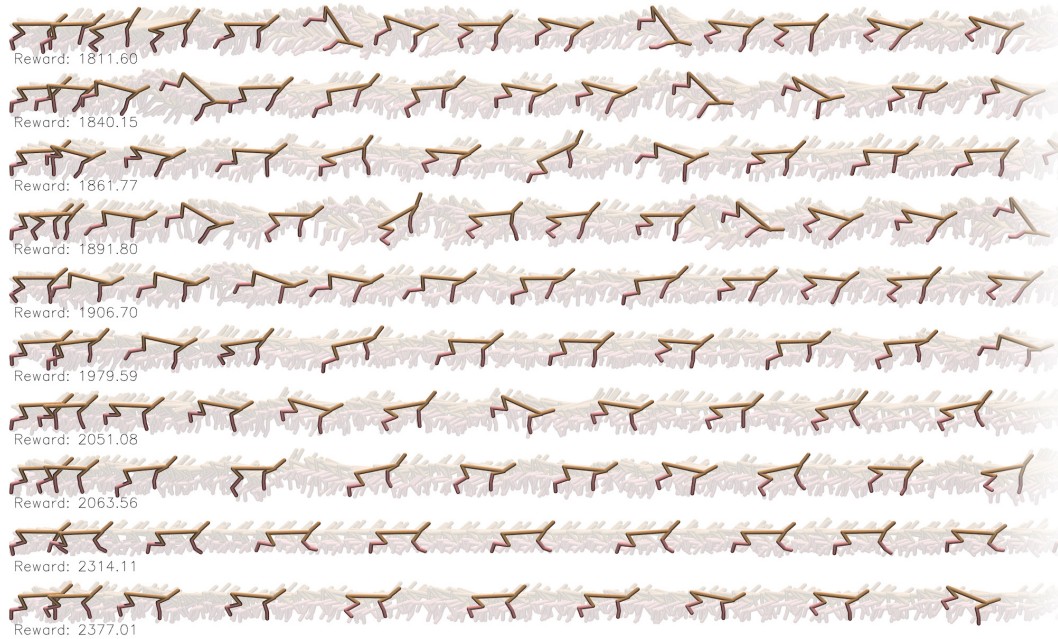

Figure 10: The visualization of policy behaviors of agents trained by our method in HalfCheetah-v3 environment. Our agents run much faster compared to PPO agents and at the mean time several patterns of motion have emerged.

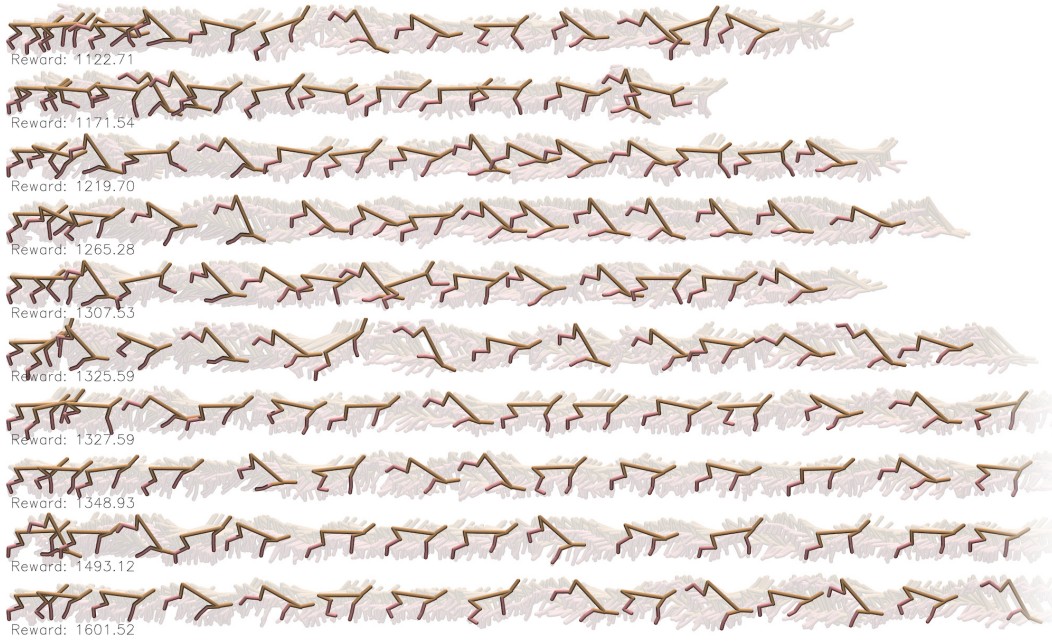

Figure 11: The visualization of policy behaviors of agents trained by PPO in HalfCheetah-v3 environment. Since we only draw fixed number of frames in each line, in the limited time steps the PPO agents can not run enough distance to leave the range of our drawing, which shows that our agents run much faster.

