# OpenReview forum: "Novel Policy Seeking with Constrained Optimization"
_ICLR.cc/2021/Conference — Reject_

### Official Review · AnonReviewer4 · 2020-10-27
**Official Blind Review #4**

**Rating:** 6
**Confidence:** 4

**Review:**

This paper proposes a novel constrained optimization based method, to optimize the expected return as well as encourage novelty of a new policy in contrast to existing policies. By modeling the problem as a constrained optimization problem, they can avoid excessive novelty seeking effectively, which is common in existing methods which model the problem with multi-objective optimization.

To be specific, they first propose a novel metric to measure the novelty of a new policy. To estimate such metric on sampled state with dense online reward, they propose an importance-based estimator for the proposed metric. With the estimation of the novelty metric, they propose to formulate the problem as a constrained optimization problem. The novelty is constrained to be larger than certain threshold r_0. In this way, the algorithm will only encourage larger novelty when the novelty is less than r_0, therefore avoiding excessive novelty seeking which may hurt the performance. They improve TNB proposed in (Zhang et al., 2019) with CTNB, where the ∇θg term exists only when the constraint is violated. They also propose another method based on Interior Point Method. Since IPM is computationally expensive and numerically unstable, they made an adaptation to RL setting, by bounding the collected transitions in the feasible region.

Overall, the method is intuitive and reasonable. I have the following questions:

1. The first contribution of this paper, is proposing a novel metric to measure the novelty of current policy in contrast to existing policies. Why propose a novel metric? Is existing metric for measuring the novelty not good? If so, can you verify your claim in experiments?

2. The hyper-parameter r_0.  From Figure 3, we can see the different performance under different novelty thresholds r_0. The algorithm seems to be sensitive to r_0, which is of course reasonable. How did you choose r_0 for different environments? Did you consider a soft r_0 rather than a hard constraint(that is, maybe the constraint has different weight for different r_0)?

---

> ### Author Response · Authors · 2020-11-11
> **Author Responses to Reviewer 4**
>
> Thanks for your insightful comments and your precise summary of our work! Please see our responses below.
>
> Q1: The benefits of using our proposed metric.
>
> 1. Different from previous approaches using KL-Divergence in modeling policy differences [1-3], the distance measure we proposed is a distance metric with the property of identity of indiscernible, symmetry, and subadditivity.
> 2. The proposed metric can be easily calculated in practice with Gaussian policy parameterization. And do not suffer from the support disjoint problems[4] when using KL-Divergence, which is extremely important for numerical stability during training.
> 3. Another way of modeling the difference between policies is to use additional neural networks[5,6]. Compared to our proposed metric, this approach is computationally expensive and can not result in immediate (per-timestep) reward which is crucial for the learning of IPD.
>
>
> Q2: How to choose $r_0$ for different environments? And soft constraint. (Also in Review#3 Q6 and Review#2 Q4)
>
> In our experiments we use a very simple method to determine the thresholds in different tasks. We regard the average distance between a group of PPO policies as baseline, and choose a threshold around that baseline in Figure 3.
> The intuition is: the novel policies we have should be more diverse than the naive PPO policies with different random seeds.
>
> Yes, using a soft constraint may help to relax the constraints in training later policies and fixing the threshold $r_0$ might be harmful for the later policies as they need to be optimized under more constraints. In our experiments we choose to fix the $r_0$ for conciseness and we believe a search on the decay mechanism of such $r_0$ can be helpful for improving the task performance. And it is also possible to determine the $r_0$’s according to the variety of other policies at each state, i.e., consider the constraints as “getting more novelty at every timestep, relatively”. We will try this in our future research.
>
>
>
> [1] Hong, Zhang-Wei, et al. "Diversity-driven exploration strategy for deep reinforcement learning." Advances in Neural Information Processing Systems. 2018.
> [2] Schulman, John, et al. "Trust region policy optimization." International conference on machine learning. 2015.
> [3] Schulman, John, et al. "Proximal policy optimization algorithms." arXiv preprint arXiv:1707.06347 (2017).
> [4] Arjovsky, Martin, Soumith Chintala, and Léon Bottou. "Wasserstein gan." arXiv preprint arXiv:1701.07875 (2017).
> [5] Burda, Yuri, et al. "Exploration by random network distillation." arXiv preprint arXiv:1810.12894 (2018).
> [6] Zhang, Yunbo, Wenhao Yu, and Greg Turk. "Learning novel policies for tasks." arXiv preprint arXiv:1905.05252 (2019).

---

> > ### Comment · AnonReviewer4 · 2020-11-24
> > **The benefits of the proposed metric.**
> >
> > Thanks for your clear response! I can get that KL-Divergence may suffer from the support disjoint problem, which may lead to numerical instability.  I am curious about how existing works, which adopt KL-divergence as distance measure, address such numerical problem. Also, is this problem crucial in practice?
> >
> > I would like to echo Reviewer2's comments on the metric. A comparative study on the proposed metric and KL-Divergence can make your claim much more stronger.

---

> > > ### Author Response · Authors · 2020-11-24
> > > **On the Distance Metric**
> > >
> > > The main concern we do not apply KL-divergence as the distance metric is due to its un-symmetry.
> > >
> > > As pointed by reviewer#2, in the work of Hong et al. 2018, the KL-divergence is used for distance calculation.  However, all the previous work including PPO/ TRPO etc. concentrates on the temporal diversity or exploration diversity of a single policy, while our work mainly focus on diversity between policies.
> > >
> > > We will soon provide a comparison between IPD with the Wasserstein metric and using KL-divergence as Hong et al. 2018 did in their work.

---

> > > > ### Author Response · Authors · 2020-11-24
> > > > **Adding Comparison Between IPD and Hong et al. 2018**
> > > >
> > > > We have updated a revision version of our paper and include a comparison between IPD, WSR with Wasserstein distance, and the method of Hong et al. 2018 using KL-divergence in Figure 4 (left). The method of Hong et al. can be regarded as the WSR method with KL-divergence as a distance metric. It can be seen that the diversity of agents trained with Hong et al. have a relatively large standard deviation than other methods, showing the instability of applying such a *distance* metric(the KL-divergence is not a distance metric, rigorously.). In 3M training timesteps, the method of Hong et al. can not converge to a satisfying performance compared to its Wasserstein correspondence, WSR. Thus the superiority of Wasserstein distance over KL-divergence can be seen clearly.
> > > >
> > > > We want to highlight that applying KL-divergence may also lead to numerical instability due to the log operation over proportions of two policy distributions, and normally a clip function[1] is used to tackle such an issue. On the other hand, using the Wasserstein distance will not suffer from such instability as the computation of the Wasserstein distance between two Gaussian distributions is much simpler.
> > > >
> > > > To sum up, we believe there are adequate reasons to use the Wasserstein metric rather than the KL-divergence:
> > > > 1. Rigorous definition of a distance metric: as we have shown in our work, using Wasserstein can help to build up the distance metric between policies.
> > > > 2. Numerical stability: the computation of Wasserstein distance between Gaussian policies is much simpler than applying the KL-divergence. And avoiding the division as well as the clip operation can improve the stability and accuracy, regarding the clipping as an approximation.
> > > > 3. Our additional experiment results in Ant-v3 support our claim that the Wasserstein distance is a better choice.
> > > >
> > > > Thank you so much for your insightful comments, reply, and advice!
> > > >
> > > > [1] Dhariwal, Prafulla, et al. "Openai baselines." (2017).

---

### Official Review · AnonReviewer3 · 2020-10-28
**Seeking novel policy only after finding a good one**

**Rating:** 4
**Confidence:** 4

**Review:**

This paper aims at novel policy seeking which incorporates curiosity-driven exploration for better reinforcement learning. This paper first propose to use  a Wasserstein-based metric to calculate the difference between policies, and use it to define  the policy novelty. With these, the authors modeled the novel policy seeking as a constrained markov decision process(CMDP) and solved it using CTNB and IPD.

1. This paper allows to consider the novelity issue dynamically.  However, when training policy according to the proposed CTNB or IPD, there should be some pretrained policies as perconditions, in other words, the proposed method needs some prior knowledge rather than learning policy from scratch. This may be a limitation for its application.

2. About the proposition 2, the single trajectory estimation is unbiased, however, the variance seems to be large, the influence about the estimation variance should be considered.

3.  in fomula (5) and (6), is $r_{int, t}$ equal to $r_{int}$ ? If is,  why use t? and what does moving average mean since there are several kinds of moving averages?

4. in fomula (4) and (6), Are Ts the same?

5. Fig.2  shows that in Waklker2d and HalfCheetah, the proposed CTNB has less novel than PPO, which doesn't match the purpose of CTNB.

6. It seems not easy to tune the novelty threshold for different task, as it performs different on different tasks. Can the author provide some insight on how to tune this.

7. Five random seeds is not sufficient for experiments.

---

> ### Author Response · Authors · 2020-11-11
> **Author Responses to Reviewer 3**
>
> Thanks for your insightful comments. Please see our responses below.
>
> Q1: Prior knowledge:
>
> No, in our work we do not assume any prior knowledge. For the problem setting, we must have a set of policies to make comparison, or we can not define the concept of novelty.
> During learning, every policy is trained from scratch.
>
> Q2: Variance of the estimation:
>
> We do not consider the variance reduction issue because in practice we find the method performs well with the single trajectory estimation, which is the most concise and convenient approach based on the proposed novelty metric.
>
> Q3: $r_{int}$, and moving average:
>
> We assume your question is on $r_{int, t}$ and $g_{int}$ because we do not use $r_{int}$ in (5) or (6).
>
> As we defined in the paper, $g_{int} = \sum_{t=0}^{t = T} r_{int, t}$ is the intrinsic reward (i.e., novelty in our problem setting) in an episode, while $r_{int, t}$ is the intrinsic reward at timestep $t$. At each timestep, the policy receives an immediate reward, and we denote different values of such rewards with subscript $t$.
>
> The moving average means we permit a new policy to perform as previous policies in some certain step. On the contrary, if we do not include such a moving average operation, a new policy should be different from previous policies in EVERY timestep, which we believe is too strong to be a constraint. For example, in locomotion tasks there might be some crucial states that only limited choice of actions can result in a healthy state in the future, and forcing a policy to perform differently in such states will strongly hinder the performance.
>
>
> Q4: Ts in Equation (4) and Equation (6):
>
> $T$’s have the same meaning of denoting the maximal environment steps. In Mujoco tasks, the default setting is 1000 and we use the default settings in our experiments.
>
> Q5: CTNB is not novel enough:
>
> While we propose two algorithms in the work, the main purpose of CTNB is to demonstrate the priority of constrained optimization perspective for the task of novelty seeking over the previous multi-objective optimization methods. In all experiments, the CTNB outperforms TNB in the primal task reward while diminishing some diversity, which supports our claim that the constrained optimization perspective can help to improve the primal task performance.
>
> As pointed, in experiments we also find that TNB can not always find more diverse policies, partially because gradient fusion in TNB and CTNB are not stable [1, 2]. On the contrary, the method of IPD does not need gradient fusions and performs more robust in different environments. According to the stronger empirical performance, IPD is the recommended solution for novelty seeking tasks.
>
> Q6: Selection of $r_0$:
>
> In our experiments we use a very simple method to determine the thresholds in different tasks. We regard the average distance between a group of PPO policies as baseline, and choose a threshold around that baseline in Figure 3.
> The intuition is: the novel policies we have should be more diverse than the naive PPO policies with different random seeds.
>
> Q7: Number of random seeds:
>
> While it is reasonable to use more random seeds to test an algorithm, in most cases, the conclusions can be drawn clearly with around five to ten seeds[3-9]. (Those references include both on-policy[7] methods and off-policy[3-6,9] methods, both continuous control[3-8] and discrete domains[9], and both model-based[8] and model-free[3-7,9] approaches in RL).
>
> Moreover, in our experimental setting, 5 random seeds can generate 5 * 10 policies for each environment in total. And it can be seen that in our main results Figure 2, conclusions can be made clearly that IPD generates more diverse policies, while at least able to keep the primal task performance.

---

> > ### Author Response · Authors · 2020-11-11
> > **References**
> >
> > [1] Sener, Ozan, and Vladlen Koltun. "Multi-task learning as multi-objective optimization." Advances in Neural Information Processing Systems. 2018.
> > [2] Yu, Tianhe, et al. "Gradient surgery for multi-task learning." arXiv preprint arXiv:2001.06782. 2020.
> > [3] Fujimoto, Scott, Herke Van Hoof, and David Meger. "Addressing function approximation error in actor-critic methods." arXiv preprint arXiv:1802.09477 (2018).
> > [4] Haarnoja, Tuomas, et al. "Soft actor-critic: Off-policy maximum entropy deep reinforcement learning with a stochastic actor." arXiv preprint arXiv:1801.01290 (2018).
> > [5] Haarnoja, Tuomas, et al. "Soft actor-critic algorithms and applications." arXiv preprint arXiv:1812.05905 (2018).
> > [6] Ciosek, Kamil, et al. "Better exploration with optimistic actor critic." Advances in Neural Information Processing Systems. 2019.
> > [7] Tessler, Chen, Guy Tennenholtz, and Shie Mannor. "Distributional policy optimization: An alternative approach for continuous control." Advances in Neural Information Processing Systems. 2019.
> > [8] Hafner, Danijar, et al. "Dream to control: Learning behaviors by latent imagination." arXiv preprint arXiv:1912.01603 (2019).
> > [9] Badia, Adrià Puigdomènech, et al. "Agent57: Outperforming the atari human benchmark." arXiv preprint arXiv:2003.13350 (2020).

---

> > > ### Author Response · Authors · 2020-11-23
> > > **Additional Experiments and More on Q5**
> > >
> > > We run additional experiments on the Ant-v3 environment to show the performance of our algorithm on the more complicated continuous control task (with 111-dim state space and 8-dim action space). Results are shown in Figure 4. Our method of IPD achieves on-par performance with PPO but improves the novelty between policies by $20$ percent.
> > >
> > > Notably, both the performance and novelty of CTNB in Ant-v3 is better than its multi-objective optimization counterpart, the TNB, we attribute the reason to the limited training timesteps in TNB and CTNB: in limited training timesteps (3M timesteps in Ant-v3), the policies trained with CTNB and TNB can not converge to well-performing policies, and therefore the behavior difference between those policies are limited (even less than PPO). On the contrary, the method of IPD does not fuse the gradient of primal task reward and the novelty reward, thus similar learning efficiency can be achieved and result in well-performing policies.
> > >
> > > This experiment demonstrates our claim on the superiority of constrained optimization perspectives of novelty-seeking again: too much pursuance of the novelty will hinder the primal task performance as well as hinder the generating of both diverse and well-performing policies.

---

### Official Review · AnonReviewer2 · 2020-10-28
**Interesting method but can be more convincing**

**Rating:** 6
**Confidence:** 3

**Review:**

Summary: This paper proposed a method to leverage the constrained optimization for policy training to learn diverse policies given some references. Based on a diversity metric defined on policy divergences, the paper employs two constrained optimization techniques for this problem with some modifications. Experiments on mujoco environments suggest that the proposed algorithms can beat existing diversity-driven policy optimization methods to learn both better and novel policies. Generally, the paper is well-written and easy to follow. Some concerns/comments:

* The state distributions of proposed CTNB and IPD are different: In the CTNB method, the trajectories will keep rollout until they reach some termination conditions such as time limit or failure behavior. However, in the IPD method, if the cumulative novel reward is below some thresholds, then the trajectories will be truncated. It will be helpful to compare the CTNB with that extra termination condition.

* Using the divergence of policies to quantify the difference between policies seems not a very innovative metric. Some related work could be:

Hong, Z. W., Shann, T. Y., Su, S. Y., Chang, Y. H., Fu, T. J., & Lee, C. Y. (2018). Diversity-driven exploration strategy for deep reinforcement learning.

It will be great if the authors can compare and explain the relationship between the proposed metric and some related ones.

* The experiments can be more convincing if more locomotion environments are included, especially some higher-dimensional environments such as Humanoid and HumanoidStandup. Also, some other environments with a long-term/sparse reward setting can be more illustrative such as some mazes or Atari games. For some of those games, since it is stage-based, the IPD might terminate some rollouts if all reasonable policies are similar at the beginning of the trajectory. For a maze example, all good policies should choose to open the door at the beginning and then behave diversely.

Other/Minor Comments:

* The choice of r_0 can affect the performance: When sequentially training the policy, should r_0 be adjusted when training each new policy?

* It can be more interesting if some visualization of hopper policy diversity is included.

---

> ### Author Response · Authors · 2020-11-11
> **Author Responses to Reviewer 2**
>
> Thanks for your insightful comments. Please see our responses below.
>
> Q1: The combination of CTNB and IPD
>
> A combination of the two proposed methods is not included in our work as we propose CTNB mainly to demonstrate the benefit of constrained optimization perspective of novelty-seeking over the previous multi-objective optimization methods. In all experiments, the CTNB outperforms TNB in the primal task reward while diminishing some diversity. However, in experiments, we also find that TNB can not always find more diverse policies, partially because gradient fusion in TNB and CTNB is not stable [1, 2]. On the contrary, the method of IPD does not need gradient fusions and performs more robust in different environments.
>
>
> Q2: On the proposed novelty metric.
>
> In previous work, the distance of policies has been explored. e.g., In TRPO and PPO, the distance between new and old policies is used to constrain the policy updates during policy learning.
> However, in these work the distances are always modeled by the KL-divergence, as in their problem settings there is no need for a rigorous distance metric, which should have the property of symmetry. In our work, the absolute values of such a distance are important and with our definition, the distance between $\theta_i$ and $\theta_j$ is the same as between $\theta_j$ and $\theta_i$. This makes our definition necessary and distinguished from the previously defined distance.
>
> In another previous work of TNB, the authors considered using Auto-Encoders as a black-box distance measure, and the evaluation of policy distances is conducted on the trajectory level. With our proposed metric, there is no need for building another neural network for novelty measurement and more importantly, the distance can be calculated at every timestep, and such immediate novelty reward (instead of episodic novelty reward) enables our implementation of IPD.
>
>
> Q3: More environments:
>
> We will run more higher-dimensional experiments and provide the results. (However, the process may take some time as the problem setting needs to train novel policies sequentially)
>
> For the maze environment, we provide a simple maze (without obstacles) environment in our appendix to visualize different methods. For more complex maze environments, considering diverse policies may not be helpful as the optimal solution can easily be defined according to the reward function. Exceptions include in some tasks, a vanilla algorithm like PPO may always converge to a bad local minimum [3]. It becomes another task of better exploration with diversity-seeking. In such cases, the criteria should be finding one policy that can solve the task instead of a set of diverse policies that are able to solve the same task in different ways, which is not the main topic of our work.
>
>
> Q4: On the threshold $r_0$:
>
> In principle, fixing the threshold $r_0$ might be harmful to the later policies as they need to be optimized under more constraints. In our experiments, we choose to fix the $r_0$ for conciseness and we believe a search on the decay mechanism of such $r_0$ can be helpful for improving the task performance.
>
> Q5: Visualization:
> We provide visualization results of the locomotion tasks in our revised paper.
>
> [1] Sener, Ozan, and Vladlen Koltun. "Multi-task learning as multi-objective optimization." Advances in Neural Information Processing Systems. 2018.
> [2] Yu, Tianhe, et al. "Gradient surgery for multi-task learning." arXiv preprint arXiv:2001.06782. 2020.
> [3] Zhang, Yunbo, Wenhao Yu, and Greg Turk. "Learning novel policies for tasks." arXiv preprint arXiv:1905.05252 (2019).

---

### Official Review · AnonReviewer1 · 2020-10-30
**Novel Policy Seeking with Constrained Optimization**

**Rating:** 4
**Confidence:** 4

**Review:**

Summary:

This paper attempts to solve the problem of seeking novel policies in reinforcement learning from a constrained optimization perspective. This new perspective motives two new algorithms to solve the optimization problem, which are based on feasible direction and the interior point methods. The authors provide empirical results on several Mujoco benchmarks.

Details:

The idea of formulating the problem from a constrained optimization perspective is interesting. This new perspective motivates new and better algorithms to solve the optimization problem.

However, I feel like the presentation is poor and the writing should be improved.

What’s the exact problem setting? The authors should clearly describe the problem setting before presenting the methods, even one paragraph would be helpful.

A lot of algorithm details are missing:

Q1. $\bar{D}^q_W (\theta_i, \theta_j)$ is a metric for any state distribution $q$. What’s the motivation of using $q = \bar{\rho}$?

Q2. When computing the policy distance, what is $\rho_{\theta_i}$ in (4)? Is it the current policy, or a reference policy?

Q3. I assume $\theta_i$ is the current policy. According to (4), the algorithm uses $\theta_i$ to get samples, and compute an importance correction ratio $q/\rho_{\theta_i}$ to approximate the distance. How is the $q(s)=\bar{\rho}(s)$ computed? The authors propose to approximate $\rho_{\theta}$ using monte-carlo methods. Does it mean the algorithm need to approximate $\bar{\rho}(s)$ using the reference policies for each $s\sim \rho_{\theta_i}$? Is there a computation issue?

Q4. This goes back to Q1. Why just using the on policy samples to estimate the distance? Is there any potential advantage to use $q = \bar{\rho}$?

Q5. Learning the stationary distribution is a hard research problem itself. See recent work for example:

Zhang, R., Dai, B., Li, L. and Schuurmans, D., 2019, September. GenDICE: Generalized Offline Estimation of Stationary Values. In International Conference on Learning Representations.

I agree the stationary distribution can be approximated using MC methods, but it might need a lot of samples as the variance is very high. This makes me wonder how is the algorithm implemented in practice, and how does the stationary distribution estimation subroutine affect the algorithm’s performance.

Other suggestions:

If I understand correctly, this paper tries to solve the problem of finding a set of novel polices that solve a given task while exhibiting different behaviors. This seems also related to the exploration problem, as some works try to make the current policy different with previous policies to encourage exploration. See for example:

Hazan, E., Kakade, S., Singh, K. and Van Soest, A., 2019, May. Provably efficient maximum entropy exploration. In International Conference on Machine Learning (pp. 2681-2691).

It might be worth to discuss how the novel policy seeking problem is related to the exploration problem.

---

> ### Author Response · Authors · 2020-11-11
> **Author Responses to Reviewer 1**
>
> Thanks for your insightful comments. Please see our responses below.
>
> [Problem Setting] What’s the exact problem setting? The authors should clearly describe the problem setting before presenting the methods, even one paragraph would be helpful.
>
> Thanks for pointing it out. Our problem setting is: given a task, we hope to find a set of diverse policies that can make a good trade-off between diversity (novelty) and the primal task performance. We regard the diversity of 10 independently trained PPO policies as baseline, and during training new policies sequentially, we aim at generating policies that behave differently from all previous policies, but try to keep the primal task performance. We will provide a more detailed description in the revision.
>
> Q1 and Q4: Why use $q = \bar{\rho}$? Why just using the on policy samples to estimate the distance?
>
> In principle, q can be selected as any distribution on the state space. It must remain unchanged in considering any given policies $\theta_i, \theta_j$ in calculating their distance according to the symmetry property of a metric by definition.
>
> However in practice, it will be difficult to sample from a given distribution other than the state visitation distribution of the current policy [1, 2]. We therefore rely on the assumption that policies generated by the same algorithm like PPO are similar in state visitation frequency and use approximation $\bar{\rho} \approx \bar{\rho_{\theta}}$ in distribution, $\forall \theta \in \Theta_{PPO}$.
> We provide some illustration on such a choice below the Equation (4) in our paper.
>
> The choice of using on-policy samples in estimating the distance is by the consideration of computational efficiency. Otherwise we may need to maintain a buffer of states that has been visited by previous policies and do much more inference in calculating the difference. Notice that the above approximation will be bad only when the visitation frequency of the current policy is clearly different from previous policies, in which case we have found a novel policy.
>
>
> Q2: what is $\rho_{\theta_i}$ in (4)? Is it the current policy, or a reference policy?
>
> $\theta_i$ is the current policy and $\theta_j$ is the reference policy.
> We have a description of this in our paper (the paragraph above Proposition 2): ‘However, during training, θi is a policy to be optimized and θj ∈ Θref is a fixed reference policy.’
>
> Q3 and Q5: Computation issue, and implementation?
>
> We believe this question can partially be answered by the response for Q1 and Q4. The importance correction is assumed to be close to $1$ as an approximation considering the computational efficiency.
>
>
> Q6: How the novel policy seeking problem is related to the exploration problem.
>
> Indeed we believe the topic of novel policy seeking is closely related to the exploration problems in RL. We have provided discussion on the relation as well as the differences between the objective and previous exploration papers in the related work.
> Generally speaking, our work provides an approach of generating a set of well-performing diverse policies, which can be potentially used for downstream applications like policy ensemble [3-5], or maybe replacing the random initialization-based bootstrapping value estimations in [6, 7]. We leave these applications of learning novel policies to future work and focus on how to get a set of well-performing diverse policies in this work.
>
> [1] Sutton, Richard S., et al. "Policy gradient methods for reinforcement learning with function approximation." Advances in neural information processing systems. 2000.
> [2] Silver, David, et al. "Deterministic policy gradient algorithms." 2014.
> [3] Anschel, Oron, Nir Baram, and Nahum Shimkin. "Averaged-dqn: Variance reduction and stabilization for deep reinforcement learning." International Conference on Machine Learning. PMLR, 2017.
> [4] Wiering, Marco A., and Hado Van Hasselt. "Ensemble algorithms in reinforcement learning." IEEE Transactions on Systems, Man, and Cybernetics, Part B (Cybernetics) 38.4 (2008): 930-936.
> [5] Faußer, Stefan, and Friedhelm Schwenker. "Neural network ensembles in reinforcement learning." Neural Processing Letters 41.1 (2015): 55-69.
> [6] Osband, Ian, John Aslanides, and Albin Cassirer. "Randomized prior functions for deep reinforcement learning." Advances in Neural Information Processing Systems. 2018.
> [7] Osband, Ian, et al. "Deep exploration via bootstrapped DQN." Advances in neural information processing systems. 2016.

---

### Author Response · Authors · 2020-11-13
**Rebuttal Revision Version Updated**

Dear Reviewers, AC, PC, and Readers,

In our revised version of the work, we
1. include a more complex environment, the Ant-v3 (in Appendix F), which has 111-dim state space and 8-dim action space, to demonstrate our claims in the paper: the constrained optimization perspective of novelty-seeking problems can help to generate well-performing policies with diverse behaviors.
2. include a paragraph of problem setting to address the concern of Review #1.
3. include more qualitative analysis in Appendix G.
4. provide experiment results in Ant-v3 (in Appendix F) to support our claim on the superiority of Wasserstein distance over KL-divergence as a distance metric in quantifying the differences between policies.

Thanks for your attention,
Paper 1214 Authors

---

### Decision · Program_Chairs · 2021-01-07
**Final Decision**

**Decision:**

Reject

**Comment:**

This paper investigate the interesting problem of policy seeking in reinforcement learning via constrained optimization. Conditioned on reviewers' judgements, this is a good submission but hasn't reached the bar of ICLR.